# U.S. regional differences in physical distancing: Evaluating racial and socioeconomic divides during the COVID-19 pandemic

Emma Zang[1]*, Jessica West[2], Nathan Kim[1], Christina Pao[3]

**1** Department of Sociology, Yale University, New Haven, Connecticut, United States of America, **2** Center for the Study of Aging and Human Development, Duke University, Durham, North Carolina, United States of America, **3** Department of Sociology, University of Oxford, Oxford, United Kingdom

\* emma.zang@yale.edu

## Abstract

Health varies by U.S. region of residence. Despite regional heterogeneity in the outbreak of COVID-19, regional differences in physical distancing behaviors over time are relatively unknown. This study examines regional variation in physical distancing trends during the COVID-19 pandemic and investigates variation by race and socioeconomic status (SES) within regions. Data from the 2015–2019 five-year American Community Survey were matched with anonymized location pings data from over 20 million mobile devices (Safe-Graph, Inc.) at the Census block group level. We visually present trends in the stay-at-home proportion by Census region, race, and SES throughout 2020 and conduct regression analyses to examine these patterns. From March to December, the stay-at-home proportion was highest in the Northeast (0.25 in March to 0.35 in December) and lowest in the South (0.24 to 0.30). Across all regions, the stay-at-home proportion was higher in block groups with a higher percentage of Blacks, as Blacks disproportionately live in urban areas where stay-at-home rates were higher (0.009 [CI: 0.008, 0.009]). In the South, West, and Midwest, higher-SES block groups stayed home at the lowest rates pre-pandemic; however, this trend reversed throughout March before converging in the months following. In the Northeast, lower-SES block groups stayed home at comparable rates to higher-SES block groups during the height of the pandemic but diverged in the months following. Differences in physical distancing behaviors exist across U.S. regions, with a pronounced Southern and rural disadvantage. Results can be used to guide reopening and COVID-19 mitigation plans.

## Introduction

In the first half of 2020, the U.S. comprised 4% of the world's population but ¼ of the confirmed coronavirus disease 2019 (COVID-19) cases and deaths [1]. By April 26, 2021, the U.S. had recorded over 31 million cases of and 569,272 deaths due to COVID-19 [2]. In an effort to contain the spread of the virus, reducing contact between infected and susceptible individuals

**Data Availability Statement:** While our demographic variables are public use and we have made them freely available here (https://github.com/18kimn/safegraph-covid-shareable), the main

variables of interest in our project come from the Safegraph Social Distancing dataset under a proprietary license. We have access to the data under an agreement pertaining to Section 11 of the Safegraph Terms of Service (found at https://shop.safegraph.com/terms-of-service/), and Safegraph is highly willing to share these variables at currently no cost to academic users, but we do not have permission to make these data public at this time. Interested researchers can request access for these data at https://www.safegraph.com/academics. The data underlying the results presented in the study are available from the SafeGraph Inc (https://www.safegraph.com/).

**Funding:** EZ received support from the National Institute on Aging (R21AG074238-01), the Research Education Core of the Claude D. Pepper Older Americans Independence Center at Yale School of Medicine (P30AG021342), the Panel Study of Income Dynamics Small Grants for Research Using Data from CDS and TAS (R25-HD083146), and the Institution for Social and Policy Studies at Yale University. JW is supported by the Duke Aging Center Postdoctoral Research Training Grant (NIA T32 AG000029). The funders had no role in study design, data collection and analysis, decision to publish, or preparation of the manuscript.

**Competing interests:** The authors have declared that no competing interests exist.

became the key strategy to prevent disease transmission [3, 4]. As such, in March 2020, public health and government officials imposed restrictions on domestic and international travel at the federal level, and began recommending physical distancing behaviors at the state level (e.g., maintaining six feet of distance, avoiding group gatherings, stay-at-home-orders, and the closure of non-essential businesses and schools [5, 6]) to mitigate the spread of COVID-19 (note that we use "physical distancing" rather than "social distancing" to emphasize the importance of preserving social—while reducing physical—interactions [5]).

While the virus has reached most areas of the U.S., there is evidence of variation across Census regions within the country. While the pandemic initially struck the West and the Northeast hardest, COVID-19 cases later rose in the Midwest and the South [2], resulting in differences in COVID-19 outcomes such as cases and deaths across Census regions [7, 8]. For example, research has revealed regional differences by the percentage of counties that met criteria for being a "hotspot." Specifically, in March-April, counties in the Northeast Census region met hotspot criteria more often than all other regions; however, by June-July, counties in the South and West Census regions were proportionally meeting hotspot criteria more than the Northeast and Midwest [9]. These differences may be due to regional variation in factors that influence COVID-19 risks, including various individual (e.g., age, pre-existing health conditions), household (e.g., poverty, household size), and community (e.g., presence of group quarters such as correctional facilities or nursing homes) factors [10]. This is consistent with a long line of research on regional variation in health and mortality [11–16].

Despite apparent regional differences in COVID-19 outcomes, there has been little focus on how physical distancing behavior varies by Census region, which is an important omission because physical distancing trends are predictive of later COVID-19 outcomes [3]. There are many reasons to believe that physical distancing may be structurally constrained by Census region. For example, racial residential segregation is more prevalent in the Northeast, Midwest, and South [17], and can constrain access to quality grocery stores with decent food supplies [18], which can lead to more frequent grocery store visits and greater potential for COVID-19 exposure. Compared to other regions, the South had a higher prevalence of poor health and pre-existing chronic conditions [16, 19], which might have increased medical facility visitation and limited physical distancing ability. Moreover, the prevalence of physical activities was particularly low in the South [19], and places with more health-protective behaviors prior to the pandemic (e.g., greater physical activity) exhibited a greater reduction in movement outside of the home [20]. The South also has higher poverty rates compared to other regions [21], and socioeconomic status (SES) is positively associated with physical distancing [22]. Understanding regional differences in physical distancing trends may highlight particular regions where COVID-19 mitigation policies and outreach should be targeted.

Additionally, region may shape SES and racial differences in physical distancing, making it crucial to examine these differences within each region. Physical distancing necessitates the ability to work from home, distance while working from home, take (un)paid time off, etc. [23, 24]. While physical distancing was generally high following state emergency guidelines, the intensity of distancing correlated dramatically with income [22]. Moreover, racial/ethnic minority groups disproportionately work in low-wage or essential work settings [25, 26] where COVID-19 exposure risk is high [27, 28]. Even when racial/ethnic minorities can remain home, they overwhelmingly live in places that put them at higher risk for COVID-19 [23, 29]. Because delayed testing and lack of accessible health care, public health resources, and paid leave are particularly severe in the South, physical distancing patterns may be particularly unequal across SES and racial groups in the South [30].

Since the beginning of the pandemic, there has been an increasing number of studies examining physical distancing trends in the US, and most of them used digital data, such as

geotagged social media data or mobility data from cell phone pings and navigation systems. The pandemic has accelerated the use of such data for various purposes including mapping population movement, developing models of disease transmission, and informing resource allocation [31, 32]. Such research has shown that physical distancing and stay-at-home orders have contributed to reducing the growth rate of COVID-19 [3, 33] and that measures of physical distancing (e.g., maximum travel distance, stay-at-home time, decreases in physical movement) are associated with a reduction in COVID-19 case rates [34, 35].

However, there is notable variation in adherence to physical distancing recommendations. Studies using varying scales (e.g., neighborhood, Metropolitan Statistical Areas, counties, Census tract) have shown that high SES individuals were more likely to engage in physical distancing behaviors compared to low SES individuals [3, 22, 36–38]. County-level data from Google Maps from February 16[th] through March 29[th], 2020 reveals racial/ethnic differences in physical distancing but emphasizes that non-White groups do not respond in the same manner to COVID-19 restrictions [39]. County-level shares of racial/ethnic minorities and rurality have also been associated with reduced physical distancing, but these trends vary across the pandemic [36]. There is also evidence that in states where the confirmed COVID-19 cases were increasing faster, people generally reduced their mobility more quickly [34, 40]. Borough-level social media data in New York City captured changes in human mobility patterns by different land use types (residential, parks, transportation facilities, workplaces), showing a decrease in mobility around tourism-related locations (e.g., Statue of Liberty ferry) and commercial and office buildings in Midtown Manhattan [41]. A study examining the correlation between the strictness of physical distancing policies and the spread of COVID-19 determined that the optimal level of physical distancing intervention should be at least 80% in order to reduce infection and the number of deaths [42].

Overall, there is a growing literature tracking mobility changes in the U.S. over the course of the pandemic. However, this literature is subject to some limitations. First, due to publication dates, many studies focus on specific time points in the pandemic, such as the first few months of the U.S. epidemic [22, 37, 39], with more recent studies able to capitalize on access to and availability of more data [36]. Second, while varying scales have been used to track mobility patterns, few studies have examined physical distancing patterns across Census regions. Describing patterns by Census region can reveal how physical distancing behaviors are impacted by structural constraints at a macro socio-spatial level, such as poverty and racial residential segregation [17, 21].

The current study aims to contribute to this nascent literature by using nationally representative data at the Census block group level to, 1) show regional trends in physical distancing practices over the course of the pandemic, and 2) examine differences within each region by race and SES (income, education, occupation) using visual tools and regression analyses. As such, the purpose of the study is to present descriptive patterns of regional variation in physical distancing rather than causal determinants of these behaviors. Findings from this study may help policymakers determine which regions are most affected and which communities might be most impacted within these regions.

## Materials and methods

We use anonymized location data from SafeGraph, Inc. which were collected from a representative sample [43] of over 20 million cell phones and recorded daily at the Census block group level from January 1[st], 2019 to December 31[st], 2020 (see S1 Fig for the completeness of the data at the national level). A Census block group is a geographical unit between the size of a Census tract and a Census block and typically contains between 600 to 3,000 people. Compared to

larger geographic divisions (e.g., counties), the block group aligns closely with neighborhood boundaries and is useful for studying segregated areas like cities.

For our main analyses, we represent the extent of physical distancing with a seven-day rolling average of the proportion staying completely at home. "Home" is defined as the geohash-7, or approximately a 153-meter by 153-meter area, that serves as the most common nighttime location for each device. This measure has been used to capture physical distancing in other studies [40, 44]. We also study various alternative measures [45] and results are consistent (see S2 Fig and regional-distancing.info for an interactive tool to explore these metrics over time at the state level).

Next, we match stay-at-home rates with demographic information on urbanicity, age, race, and SES from the 2015–2019 five-year American Community Survey (ACS) at the Census block group level. From the ACS, we obtain urbanicity (urban vs. rural); the proportion of residents over age 65; the proportion of the population identifying as Black of any ethnicity; median household income; proportion of Bachelor's degree holders; proportion of frontline workers (see S1 Appendix for definition of frontline workers and robustness checks); population density (people/km^2), and average commute time.

As there is no consensus about which geographic level (region, division, etc.) to use to track health differences [46], we use Census region: Northeast, Midwest, South, and West [47]. Regional analyses may mask heterogeneity across counties/states; however, regional analyses minimize migration effects (people are half as likely to move between regions as between states in any given year [48]) and issues related to classifying people who live/work in different counties/states.

## Statistical methods

We present trends in the stay-at-home proportion by Census region and conduct linear regression models at the Census block group level to examine regional differences over time. Specifically, we regress the stay-at-home proportions across Census regions, the linear and quadratic forms of the number of days since January 1st, 2020, and a "time period" variable as well as its interactions with Census regions. To account for variations from the nationwide surge in stay-at-home rates throughout April, we defined these time periods as before April 1st, April 1st–May 1st, and after May 1st. In addition to the baseline model without controls, to examine whether the observed physical distancing patterns are mainly driven by age and racial compositions, SES, and urbanicity, we control for the following covariates in the model: urbanicity, proportion of residents over age 65, the proportion of the population identifying as Black of any ethnicity, median household income, proportion of Bachelor's degree holders, proportion of frontline workers, population density (people/km^2), and accessibility which is measured using the average commute time of residents.

We then plot the physical distancing patterns by race and SES for each region. To represent time, stay-at-home rates, and demographic measures together on one plot per demographic variable, we divided the nation into deciles for each measure. Because there are many block groups with 0 Black residents, we grouped all block groups with 0 Black residents into a "0" decile, resulting in the removal of the 1st and 2nd decile and a considerably smaller sample size in the 3rd decile. We investigate all of these SES trends within each region to investigate how these social conditions' relationship with physical distancing varies by geography. For the dimension of race, considering Black Americans disproportionately reside in urban areas, we divide block groups across deciles of proportion Black and the urban-rural status of the county (based on the U.S. Census Bureau's 2010 Urban-Rural Classification System [49]). We additionally conduct linear regression models for each Census region to examine whether the differences along the racial and SES lines are statistically meaningful. For each Census region, we

use race/SES, time period, and the interaction terms between race/SES and time period to predict physical distancing, after controlling for linear and quadratic forms of days since January 1st, 2020. Besides our focuses on race and SES, we present additional results by proportions of residents over age 65 in S5 Table.

## Results

Fig 1 presents stay-at-home patterns by Census region. For contextualization purposes, we show the 10th and 40th state-level stay-at-home orders, issued in Michigan (March 24th) and South Carolina (April 7th). Stay-at-home order information was taken from Boston University's COVID-19 U.S. State Policy Database [50].

Before March, the stay-at-home proportion was similar across regions, with 24–25% of the population staying home on any given day. After the enactment of stay-at-home orders and school closures, nearly all regions experienced a sharp increase in the stay-at-home proportion. The largest increase occurred in Northeast block groups: The seven-day rolling average of the stay-at-home proportion increased from 0.25 on March 1st to 0.46 near the height on April 1st before falling to 0.35 on December 31st. In the South, this trend repeats at considerably lower rates: The rolling stay-at-home proportion rose from 0.24 (March 1st) to 0.37 (April 1st), then drops to 0.3 (December 31st). The West and Midwest fall in between, with the proportion on these dates for the Midwest being 0.25, 0.41, and 0.32 compared to 0.25, 0.43, and 0.35 for the West. These downward arching trends over time are notable given the at best, stagnating, and at worst, intensifying, case rates.

Table 1 presents OLS model results, which confirm our observations in Fig 1. All variables are predictive of stay-at-home rates with P-values less than 0.001. The left panel shows the results without race, SES, and urban status controls. There was a surge in physical distancing during April; nonetheless, this increase declined after April, though physical distancing rates were still higher than pre-pandemic rates. Before April, physical distancing rates were highest in the Northeast, followed by the West and Midwest, and were the lowest in the South. The

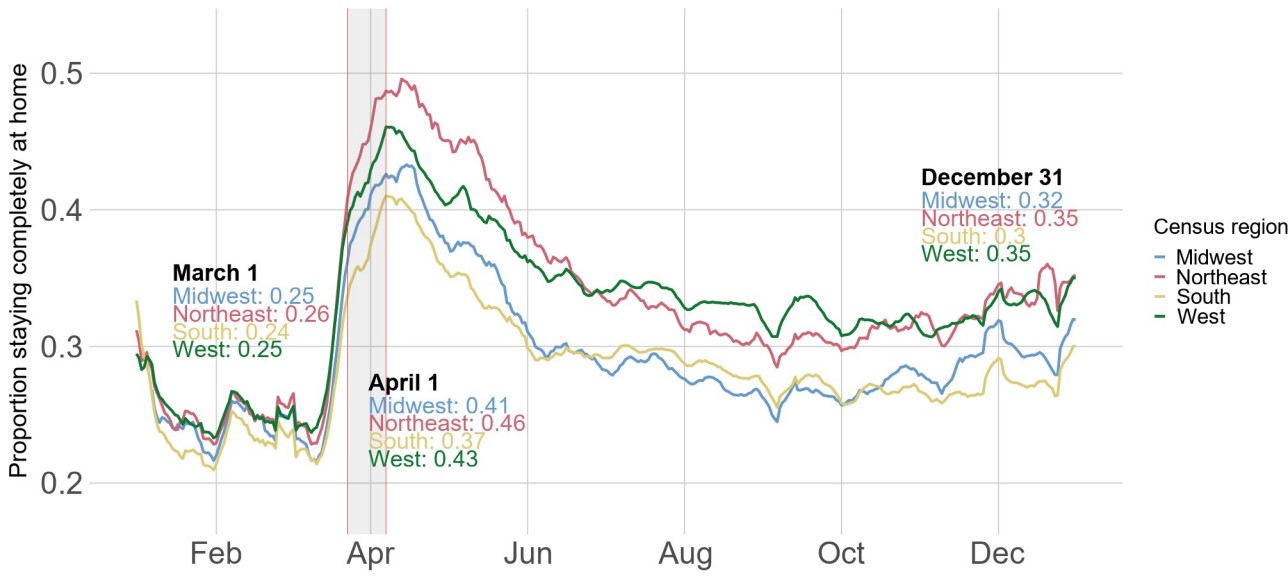

**Fig 1. Stay-at-home patterns by Census region.**

**Table 1. Regional disparities in physical distancing.**

| Variable | Without Control Variables | | | With Control Variables | | |
|---|---|---|---|---|---|---|
| | Coefficient | SE | 95% CI | Coefficient | SE | 95% CI |
| Days from January 7th | | | | | | |
| Linear term | -9.22E-06 | 7.31E-07 | (-1.07E-05, -7.79E-06) | -4.74E-06 | 7.10E-07 | (-6.14E-06, -3.35E-06) |
| Quadratic term | -3.67E-07 | 1.56E-09 | (-3.70E-07, -3.64E-07) | -3.75E-07 | 1.51E-09 | (-3.78E-07–3.72E-07) |
| Median household income (dollars) | | | | -2.16E-07 | 3.81E-10 | (-2.16E-07, -2.15E-07) |
| Proportion Black | | | | 0.049 | 4.57E-05 | (0.049,0.049) |
| Proportion frontline workers | | | | 0.033 | 8.61E-05 | (3.27E-02, 3.31E-02) |
| Proportion with a Bachelor's degree | | | | 0.053 | 7.20E-05 | (0.053, 0.053) |
| Proportion of residents over 65 | | | | 0.013 | 9.12E-05 | (0.013, 0.013) |
| Urbanicity (1 = Urban, 0 = rural) | | | | 0.036 | 2.64E-05 | (0.008, 0.009) |
| Average commute time | | | | 0.002 | 1.27E-06 | (0.002, 0.002) |
| Population density (people/km^2) | | | | -1.19E-13 | 1.92E-12 | (-3.389E-12, 3.65E-12) |
| Period (Reference = Before April 1st) | | | | | | |
| April 1st-30th | 0.144 | 8.09E-05 | (0.143,0.144) | 0.145 | 7.85E-05 | (0.144,0.145) |
| After May 1st | 0.049 | 7.19E-05 | (0.049,0.049) | 0.049 | 6.99E-05 | (0.049,0.049) |
| Region (baseline = Midwest) | | | | | | |
| Northeast | 0.012 | 5.74E-05 | (0.012,0.012) | 3.29E-04 | 5.61E-05 | (2.19E-04, 4.40E-04) |
| South | -0.016 | 4.99E-05 | (-0.016, -0.016) | -0.022 | 4.87E-05 | (-0.022, -0.022) |
| West | 0.009 | 5.58E-05 | (0.009,0.009) | 0.005 | 5.45E-05 | (0.005, 0.005) |
| Period and region interaction | | | | | | |
| April 1st-30th * Northeast | 0.054 | 1.12E-04 | (0.054,0.055) | 0.055 | 1.10E-04 | (0.055,0.055) |
| After May 1st * Northeast | 0.032 | 6.67E-05 | (0.031,0.032) | 0.031 | 6.50E-05 | (0.031,0.032) |
| April 1st-30th * South | -0.012 | 9.77E-05 | (-0.012, -0.012) | -0.012 | 9.50E-05 | (-0.012, -0.011) |
| After May 1st * South | 0.007 | 5.79E-05 | (0.007,0.007) | 0.007 | 5.64E-05 | (0.007, 0.008) |
| April 1st-30th * Midwest | 0.016 | 1.09E-04 | (0.016,0.016) | 0.016 | 1.06E-04 | (0.016,0.016) |
| After May 1st * Midwest | 0.035 | 6.48E-05 | (0.035,0.036) | 0.036 | 6.29E-05 | (0.035,0.036) |
| Intercept | 0.313 | 8.82E-05 | (0.313, 0.313) | 0.212 | 7.09E-05 | (0.212,0.212) |
| Sample size | 75,417,491 | | | 74,186,932 | | |
| Adjusted r-squared | 0.238 | | | 0.306 | | |

Note: All p-values are smaller than 0.001.

interactions between region and the time periods show that regional differences expanded during the surge in physical distancing from April 1st to 30th and contracted afterwards: in the Northeast, the difference in stay-at-home rates before April 1st and between April 1st and 30th is 0.054 [CI: 0.054, 0.055] higher than the difference in stay-at-home-rates in the Midwest, suggesting that the baseline differences in physical distancing between the Midwest and the Northeast were made even greater during the peak of physical distancing.

Results in the rightmost panel control for age, race, SES, urban status, population density, and average commute time. Proportion of college degree holders, proportion Black, proportion of frontline workers, proportion of residents over age 65, urbanicity, and average commute time positively predict physical distancing, whereas population density and median household income negatively predict physical distancing before April. The coefficients for period, region, as well as the interaction terms remain largely unchanged, indicating that adjusting for these variables does not change the general patterns of regional differences observed in Fig 1. Supplementary results (S4 Fig) by Census division and state further show considerable variations within and across Census divisions.

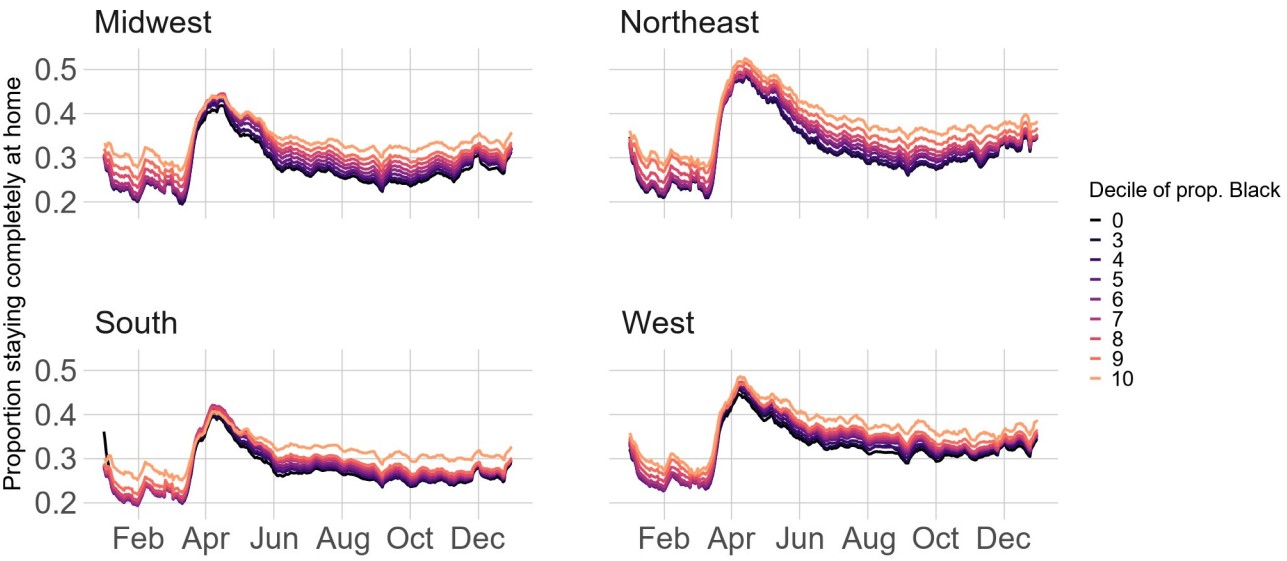

Fig 2. **Stay-at-home patterns by Census region and proportion Black.**

Fig 2 presents stay-at-home patterns by the proportion of Black residents for each Census region. Within each region, block groups with the highest proportion of Black residents had the highest stay-at-home rates throughout the pandemic, albeit to varying extents. Fig 2 also reveals that nationwide differences between the block groups with the most and least Black residents vary across regions. The difference in the seven-day rolling stay-at-home rate for block groups with the highest and lowest proportion Black residents in the Northeast is roughly 0.05 on April 1$^{st}$, 0.08 on August 1$^{st}$, and 0.05 in December. In the South, these differences are smaller: 0.01, 0.06, and 0.03, respectively. Results from regression models reinforce these trends, with the coefficients for proportion Black residents being positive across regions before April (S1 Table). The negative interaction terms between proportion Black residents and the time period of April across regions indicate that differences in physical distancing across a block's proportion of Black residents narrowed during April. These differences further narrowed after April in the Midwest and South but increased in the Northeast and West. All aforementioned results are statistically significant with P-values smaller than 0.001.

Our finding that block groups with higher proportions of Blacks tended to stay home at higher rates appears to contradict some reports on large cities (e.g., Detroit [51, 52]). Further investigation (Fig 3) explains these inconsistent findings. When looking at rural-urban divisions separately, block groups with the highest proportion of Black residents appear to stay home the *least* during the peak of the pandemic. However, urban residents generally stayed at home at a much higher rate than rural residents (Fig 3). Blacks disproportionally reside in cities, and therefore, when looking at patterns nationwide (see Fig 2), block groups with the highest proportion of Black residents stayed at home at the highest rates.

To address SES differences, we show Census region physical distancing trends by occupation (Fig 4), educational attainment (Fig 5), and median household income (Fig 6). Patterns are similar across the three variables. Pre-pandemic, higher SES block groups (e.g., lowest proportion of frontline workers, highest proportion of Bachelor's-degree-holders, or highest proportion in the top decile of median household income) stayed home at the lowest rates; however, this trend reversed throughout March in the Midwest, South, and West before

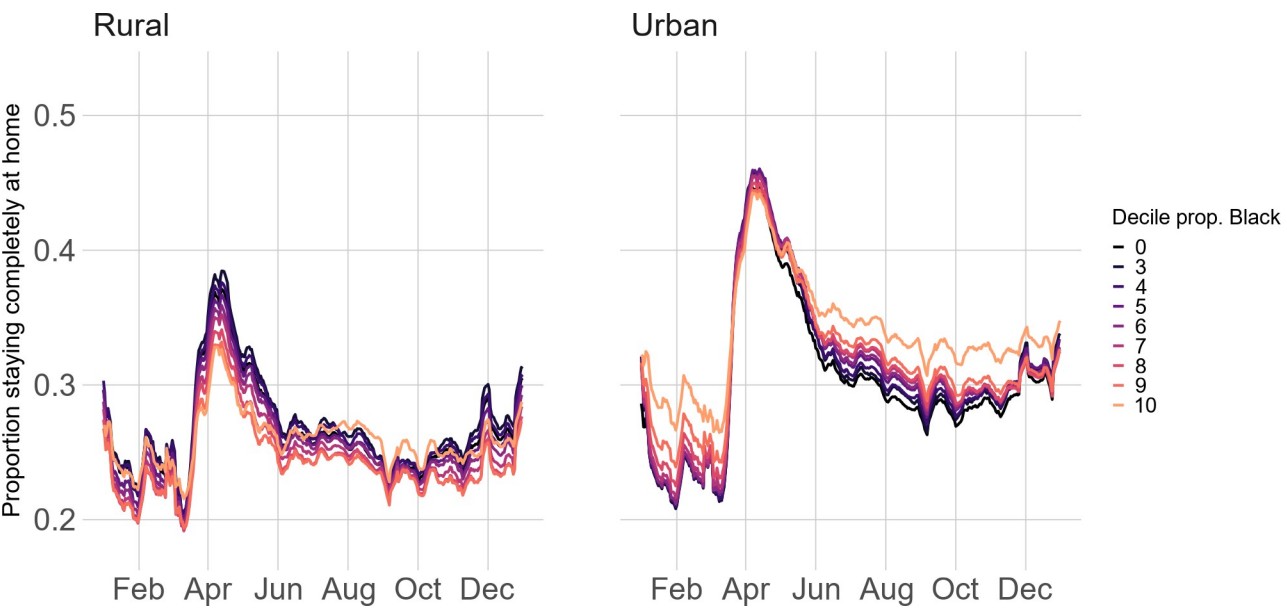

**Fig 3. Stay-at-home patterns by race across urban-rural status.**

converging in months following. In the Northeast, however, lower SES block groups stayed home at comparable rates to higher SES block groups during the height of the pandemic but diverged in the months following. For example, while the difference in stay-at-home rates between block groups with the least and most frontline workers in the South is 0.01 on April 1st, this difference is -0.05 for the Northeast. Similarly, in the South, the difference in stay-at-home rates between the most- and least-educated block groups on April 1st is 0.11; in the Northeast, the difference is considerably smaller at about 0.02. Regression results (S2–S4 Tables) confirm these results (all results statistically significant at p<0.001).

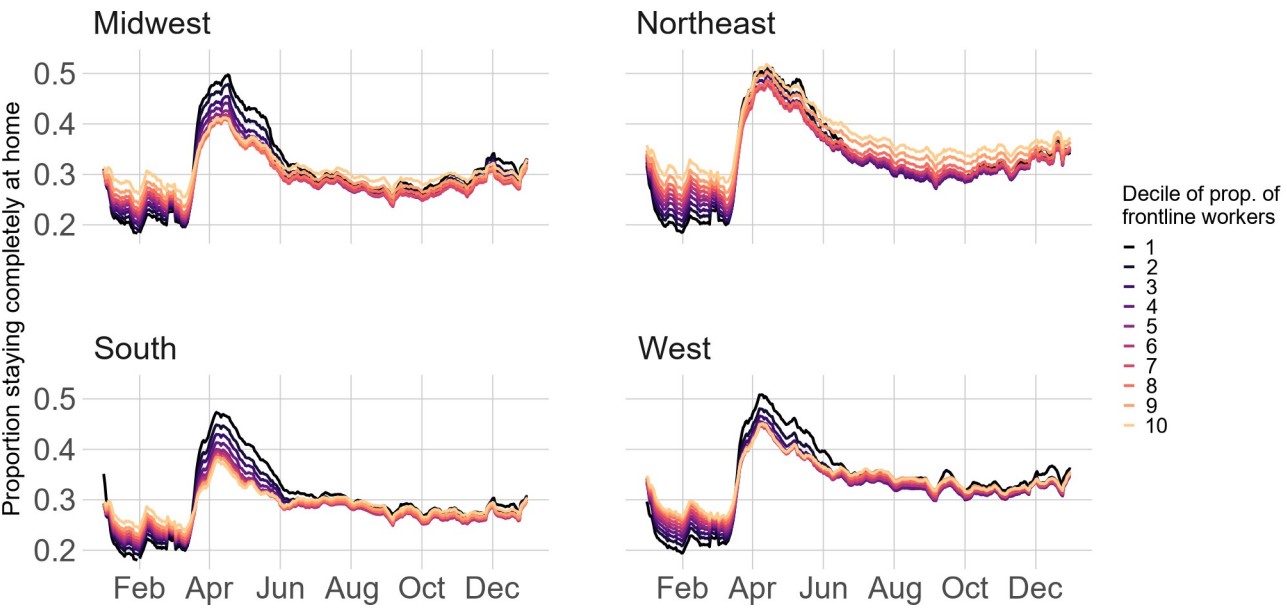

**Fig 4. Stay-at-home patterns by Census region and proportion frontline workers.**

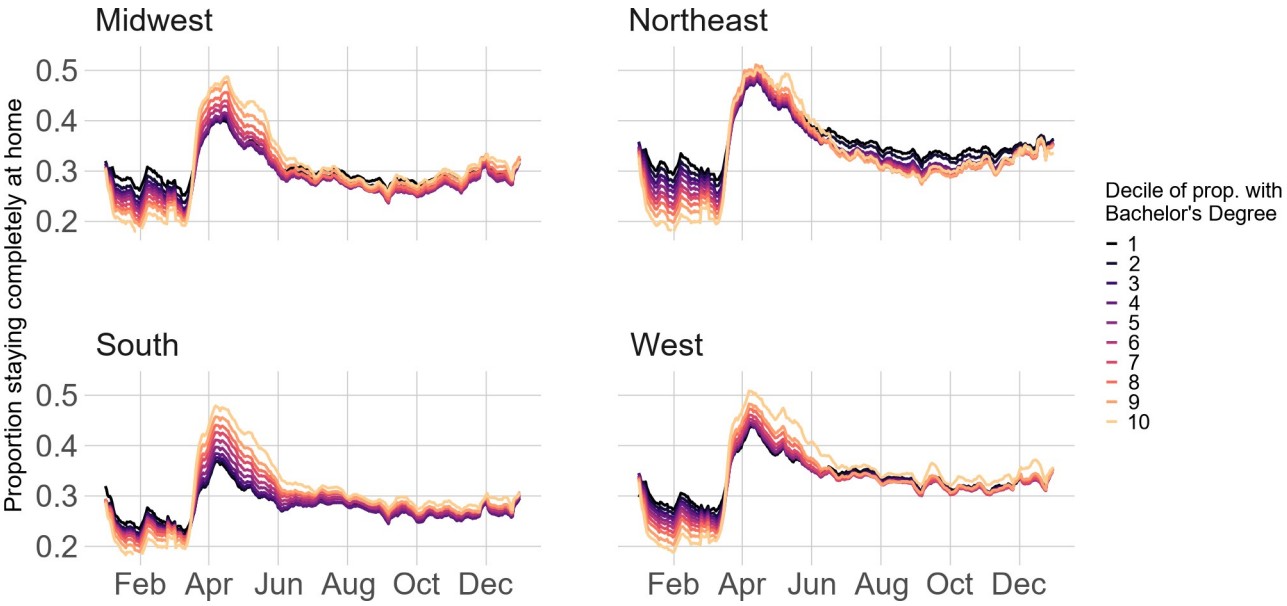

**Fig 5. Stay-at-home patterns by Census region and proportion Bachelor's degree holders.**

## Discussion

Decades of research has documented regional variation in health outcomes, life expectancy, and mortality [11–16]. Recent research indicates that regional variation has remained a crucial part of understanding health-related patterns in the U.S. during the COVID-19 pandemic. For instance, U.S. Census region was associated increases in psychological stress during the pandemic [53], interest and adoption in telehealth [54], and COVID-19 preparedness in home health agencies [55]. Research that disaggregated COVID-19 outcomes by region was also

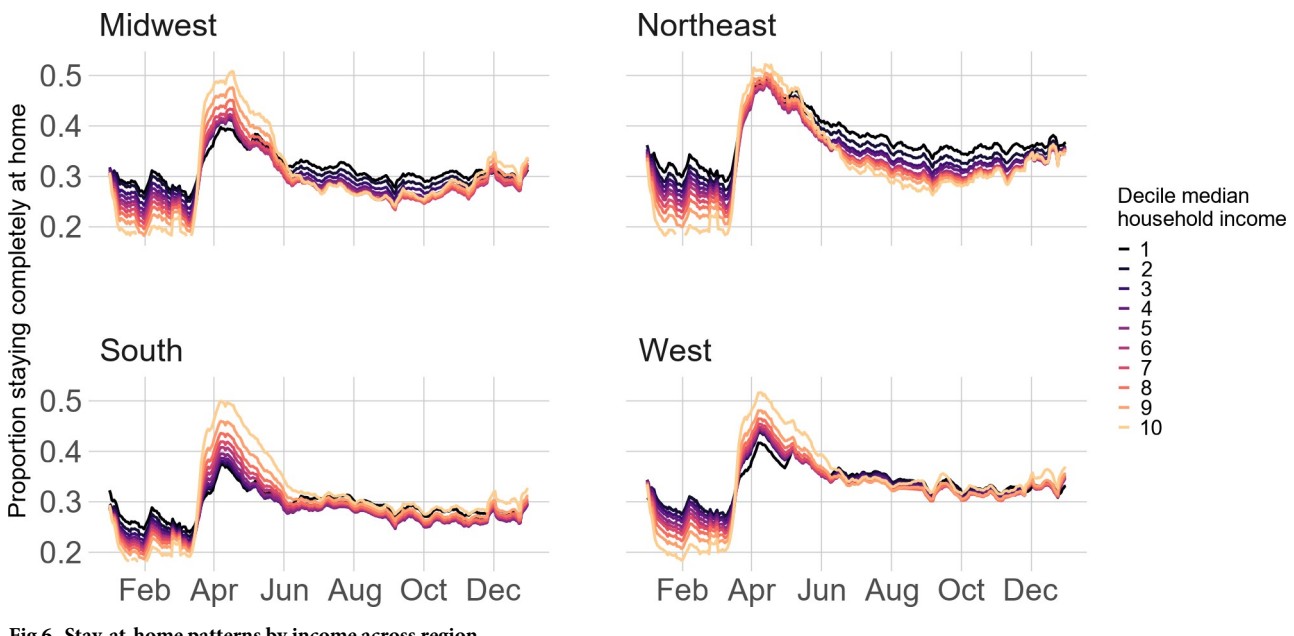

**Fig 6. Stay-at-home patterns by income across region.**

crucial in confirming the nationwide pattern of the changing age distribution (i.e., the shift in highest incidence of cases from older adults to younger adults) over the course of the pandemic [56].

However, few studies have examined regional differences in physical distancing behaviors: This is problematic because physical distancing trends are predictive of later COVID-19 outcomes [3, 4] and access to physical distancing may be structurally constrained by U.S. region. For instance, region-based research has shown that county-level factors are associated with physical distancing behaviors. Counties with more health-protective behaviors prior to the pandemic (e.g., less obesity, greater physical activity) exhibited a greater reduction in movement outside of the home compared to counties with fewer health-protective behaviors [20], and counties with higher rates of poor housing conditions (e.g., overcrowding) have more COVID-19 deaths [57]—suggesting that an inability to distance from others may explain the increased incidence of COVID-19. There is also evidence showing that high SES individuals were more likely to engage in physical distancing behaviors compared to low SES individuals [3, 22, 36–38]. Such studies provide preliminary evidence that understanding regional differences in physical distancing trends may highlight particular regions where COVID-19 mitigation policies and outreach should be targeted.

Building on this literature, the current study examines physical distancing trends across U.S. regions, and racial and SES differences within region. First, physical distancing practices vary widely across U.S. regions (see S3 and S4 Figs), with a particular disadvantage for the South. Southerners tend to lack access to health insurance [58, 59], which is likely exacerbated by pandemic-related unemployment [60]. Moreover, poverty is persistently higher in the South [21], further restricting access to the resources needed to live a healthy life. Inability to engage in physical distancing behaviors puts Southerners at greater risk for negative COVID-19 outcomes. Notably, each region stayed home at the same times even though the pandemic struck some regions much later than others. However, while all regions stayed home at the highest rates in April, the South had the lowest physical distancing rates of the four regions. For a few months (July-October), the Midwest replaced the South with the lowest physical distancing rates. Over the entire time period, our study shows that the Southern disadvantage in health and mortality [13, 16] (and, for a few months, a Midwestern disadvantage [58]) extends to physical distancing behaviors. As such, services, interventions, social safety nets, and public expenditures may be particularly necessary to help people living in the South survive the pandemic.

In addition to a Southern disadvantage, our results add to previous literature documenting a rural disadvantage. Rural America is challenged by lack of access to health care, poor health behaviors, poverty, and educational underachievement—social factors that are additionally challenging during the COVID-19 pandemic [61]. We find that rural residents are less likely to stay home compared to urban residents but note that existing media and reports tend to focus on physical distancing in cities. Thus, emphasis on physical distancing behaviors may be particularly important within rural areas.

Second, the overrepresentation of Black individuals in the number of cases, hospitalizations, and deaths associated with COVID-19 [24] is not simply driven by a difference in physical distancing patterns. In fact, nationwide, block groups with more Black residents generally stayed-at-home more than block groups with fewer Black residents. Existing media outlets and reports tended to focus only on cities, and therefore claimed that Blacks were disproportionally affected due to physical distancing patterns [51, 52]. In contrast, our results by rural-urban status and race highlight the need to study the interaction between different social conditions in creating observed stay-at-home patterns.

Third, physical distancing patterns vary across SES: physical distancing is higher among block groups that are wealthier, more educated, or contain the lowest proportion of frontline

workers. However, wealthier individuals became more mobile at the onset of summer (June/ July), likely to travel to summer destinations [62]. This socioeconomic disadvantage intersects with racial disadvantage, as demonstrated by other research that reveals higher infection rates among disadvantaged racial and socioeconomic groups due to mobility differences: individuals from disadvantaged groups are unable stay at home and the points of interest (e.g., grocery stores) that they visit are more crowded and thus associated with higher COVID-19 risk [63]. Our results confirm that SES is an important factor for COVID-19 exposure and mitigation strategies and extends research examining physical distancing and income [22] by also including measures of education and occupation.

Together, our findings reflect decades of research showing that racial and socioeconomic differences are social conditions that contribute to *health* differences [64, 65]. As a result of persisting social inequities, individuals and groups that were more likely to experience health differences prior to the pandemic are also those at highest risk for negative consequences of COVID-19 [27, 66, 67]. This heightened risk has been attributed to the numerous social, health, and environmental conditions which place racial minorities and low SES individuals at disparate risk of the negative effects of COVID-19 via poor access to medical care, (quality) health insurance, or healthy foods; inequality in education and income; living in highly segregated, disenfranchised neighborhoods with poor quality housing and greater exposure to pollution; and more [23, 24, 29, 64, 68]. Moreover, these groups disproportionately comprise the "essential" or "frontline" worker category, which limits their ability to work from home [25, 28]. Thus, the COVID-19 pandemic is highlighting deeply embedded social and structural inequities that contribute to health differences in the U.S.

## Limitations

Our results should be interpreted in light of limitations. First, within each block group there could be higher mobile phones usage for those of higher SES [32]. Low SES individuals may be unable to pay for cell phones and bills, and older individuals may not use location-transmitting cell phones [69, 70]. This may lead to an overestimation of the percentage of residents staying at home among these demographics. Relatedly, mobile phone location data may have larger errors in low-SES areas due to poor quality of GPS signals or noises. However, assuming these issues exist to a comparable extent across Census regions, our observed regional differences in physical distancing hold.

Second, our measure of physical distancing does not include other virus avoidance practices (e.g., mask-wearing; maintaining six feet of distance from others). It is possible that individuals or groups may adhere to some practices but not others; for instance, if individuals cannot stay at home, they may instead practice mask-wearing at higher rates. Examination of their stay-at-home practices would therefore be an incomplete characterization of physical distancing. Additionally, SafeGraph's definition of "home" may lead to larger measurement errors for dense urban areas where residents typically reside in small apartment buildings than for less urban areas. Moreover, the implication of staying completely at "home" for urban and rural residents may differ, considering the large difference in population density.

Third, due to data limitation, there is a temporal gap between the ACS data (2015–2019) and the physical distancing data (2020). It is possible that the racial composition and SES for some Census block groups may have changed in the past several years, and therefore the sociodemographic characteristics matched to some Census block groups may be inaccurate. However, existing studies suggest that neighborhood and/or Census block group characteristics change slowly over time or actively stabilize. Neighborhoods have been shown to have stabilizing rates of chronic poverty or persisting affluence over the past few decades [71]. Moreover, a

review of neighborhood change from the past 50 years showed that the most common pathway of neighborhood trajectories was no change at all [72].

Fourth, our preliminary analyses by Census division and state in S3 and S4 Figs suggest that the variations within each Census region may be even greater than those between the regions. Future studies should expand our analyses to further compare physical distancing patterns at the Census division or state level. Finally, this study primarily demonstrates the descriptive patterns of regional differences in physical distancing. Future studies should adopt advanced spatial modelling strategies to examine its causal determinants, such as transit accessibility, population density, and non-pharmaceutical interventions, etc., net of the spatial spillover effects from nearby regions.

## Conclusion

Results from our study can be used by policymakers and politicians to guide plans for reopening. Despite concerns regarding COVID-19-related disparities in cases, hospitalizations, and deaths, there is limited evidence on how reopening policies disparately impact society [63]. This has led to calls for research that not only identifies the determinants of these disparities, but also that proposes policy approaches to mitigate them [73, 74]. Our analysis of location data suggests that some COVID-19 differences may be avoidable if short-term policy decisions address the amount of mobility allowed. Officials in high-risk areas may choose to adopt policies that will reduce infection densities by supporting improvements in, for example, income support, paid leave policies that allow essential workers to limit their mobility when sick, access to workplace infection protection for essential workers, and access to free and available COVID-19 testing [63].

Future research should study physical distancing along the axes of social stratification that we consider here. In addition to the dimensions considered in this study, when data become available, future studies can further examine the disparities among Census block groups by other important dimensions such as political affiliations and religion. Research is needed at the individual level to account for these intersecting barriers to health and well-being, to examine physical distancing alongside personal hygiene practices (e.g., handwashing), and to ensure representativeness in a noninvasive manner. To the extent possible, approaches should combine interview, ethnography, and survey methodologies to examine physical distancing with greater nuance and thorough noninvasive practices, complementing the results from our study which used quantitative methods and a particularly large dataset.

## Supporting information

**S1 Appendix.**
(DOCX)

**S1 Fig. Stay-at-home rates on April 1, 2020.**
(PNG)

**S2 Fig. Alternative Measures of Physical Distancing.**
(TIFF)

**S3 Fig. Stay-at-home patterns by Census division.**
(TIFF)

**S4 Fig.** (A-C) Stay-at-home patterns by state.
(PNG)

**S5 Fig. Stay-at-home patterns by occupation: Expanded definition of 'frontline workers'.**
(TIFF)

**S1 Table. Physical distancing and proportion Black, by region.**
(DOCX)

**S2 Table. Physical distancing and the proportion of frontline workers, by region.**
(DOCX)

**S3 Table. Physical distancing and the proportion of Bachelor's degree holders, by region.**
(DOCX)

**S4 Table. Physical distancing and median household income, by region.**
(DOCX)

**S5 Table. Physical distancing and proportion of residents over the age of 65, by region.**
(DOCX)

## Author Contributions

**Conceptualization:** Emma Zang, Jessica West.

**Data curation:** Emma Zang, Nathan Kim.

**Formal analysis:** Nathan Kim.

**Methodology:** Emma Zang, Nathan Kim.

**Project administration:** Emma Zang.

**Resources:** Emma Zang.

**Visualization:** Nathan Kim.

**Writing – original draft:** Emma Zang, Jessica West, Christina Pao.

**Writing – review & editing:** Emma Zang, Jessica West, Christina Pao.

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
