## [Decision Letter · Decision Letter 0]

29 Jun 2021

PONE-D-21-13855

U.S. Regional Differences in Physical Distancing: Evaluating Racial and Socioeconomic Divides During the COVID-19 Pandemic

PLOS ONE

Dear Dr. Zang,

Thank you for submitting your manuscript to PLOS ONE. After careful consideration, we feel that it has merit but does not fully meet PLOS ONE’s publication criteria as it currently stands. Therefore, we invite you to submit a revised version of the manuscript that addresses the points raised during the review process.

We look forward to receiving your revised manuscript.

Kind regards,

Nickolas D. Zaller

Academic Editor

PLOS ONE

Journal Requirements:

3. Please include your tables as part of your main manuscript and remove the individual files. Please note that supplementary tables should remain uploaded) as separate "supporting information" files.

5. We note that Figure Supporting Information ' 2020-04-01.png' in your submission contain map images which may be copyrighted. All PLOS content is published under the Creative Commons Attribution License (CC BY 4.0), which means that the manuscript, images, and Supporting Information files will be freely available online, and any third party is permitted to access, download, copy, distribute, and use these materials in any way, even commercially, with proper attribution. For these reasons, we cannot publish previously copyrighted maps or satellite images created using proprietary data, such as Google software (Google Maps, Street View, and Earth). For more information, see our copyright guidelines: http://journals.plos.org/plosone/s/licenses-and-copyright.

5.1.    You may seek permission from the original copyright holder of Figure Supporting Information ' 2020-04-01.png' to publish the content specifically under the CC BY 4.0 license. 

5.2.    If you are unable to obtain permission from the original copyright holder to publish these figures under the CC BY 4.0 license or if the copyright holder’s requirements are incompatible with the CC BY 4.0 license, please either i) remove the figure or ii) supply a replacement figure that complies with the CC BY 4.0 license. Please check copyright information on all replacement figures and update the figure caption with source information. If applicable, please specify in the figure caption text when a figure is similar but not identical to the original image and is therefore for illustrative purposes only.

Reviewers' comments:

Reviewer's Responses to Questions

**Comments to the Author**

1. Is the manuscript technically sound, and do the data support the conclusions?

Reviewer #1: Yes

Reviewer #2: Yes

2. Has the statistical analysis been performed appropriately and rigorously? 

Reviewer #1: No

Reviewer #2: Yes

3. Have the authors made all data underlying the findings in their manuscript fully available?

Reviewer #1: Yes

Reviewer #2: No

4. Is the manuscript presented in an intelligible fashion and written in standard English?

Reviewer #1: Yes

Reviewer #2: No

5. Review Comments to the Author

Reviewer #1: I am at a bit of a loss with this manuscript. You can measure physical distancing at the block group level, yet you content yourself with regional-level findings. On the one hand, this does little to build on the existing discussion as other research has already explored COVID-19 distancing behaviors by region. Gibbons (2021), for example, was just published in this journal on this question using the Google Mobility data. On the other hand, and maybe more importantly, this is a huge missed opportunity! The SafeGraph data offers the potential to address much more in-deph, and interesting, questions compared to what Gibbons and others have been able to do. The level of focus could be shifted to a smaller scale like metropolitan areas or some other smaller unit, which would offer much more richness.

Given the potential for spatial bias with such an ecological dataset, were spatial methods considered? At the very least, global and local Moran’s I tests should be conducted on the outcome measures to see if they suffer from any underlying spatial autocorrelation. If autocorrelation exists, more advance spatial modelling that accounts for spatial lag or error should be considered.

There are some other predictors that should be added to the models, including population density, transit accessibility (number of commuters, commute time, car users/public transit users, etc). These kinds of measures will help account for the need and ability of people to avoid work and other places.

Gibbons, Joseph. 2021. “Distancing the Socially Distanced: Racial/Ethnic Composition’s Association with Physical Distancing in Response to COVID-19 in the U.S.” PLOS ONE 16 (5): e0251960. https://doi.org/10.1371/journal.pone.0251960.

Reviewer #2: -The manuscript topic is interesting and timely.

-To improve this manuscript:

A-The authors need to do more analysis and explanation for the their study.

B-The authors need to explain how this study is different and unique from the other studies?

C- The Manuscript writing language needs to be edited and improved.

6. PLOS authors have the option to publish the peer review history of their article (what does this mean?). If published, this will include your full peer review and any attached files.

Reviewer #1: **Yes: **Joe Gibbons

Reviewer #2: **Yes: **Talal Daghriri

---

## [Author Response · Author response to Decision Letter 0]

30 Jul 2021

Please see the response-to-reviewers file uploaded.

---

## [Decision Letter · Decision Letter 1]

23 Aug 2021

PONE-D-21-13855R1

U.S. Regional Differences in Physical Distancing: Evaluating Racial and Socioeconomic Divides During the COVID-19 Pandemic

PLOS ONE

Dear Dr. Zang,

Thank you for submitting your manuscript to PLOS ONE. After careful consideration, we feel that it has merit but does not fully meet PLOS ONE’s publication criteria as it currently stands. Therefore, we invite you to submit a revised version of the manuscript that addresses the points raised during the review process.

While you will note that the reviewers had differing opinions as to the responsiveness of your revised manuscript, one reviewer did not feel that concerns related to modeling and geographic scale were adequately addressed.  Please pay careful attention to these comments and address them in your subsequent revision. 

We look forward to receiving your revised manuscript.

Kind regards,

Nickolas D. Zaller

Academic Editor

PLOS ONE

Reviewers' comments:

Reviewer's Responses to Questions

**Comments to the Author**

1. If the authors have adequately addressed your comments raised in a previous round of review and you feel that this manuscript is now acceptable for publication, you may indicate that here to bypass the “Comments to the Author” section, enter your conflict of interest statement in the “Confidential to Editor” section, and submit your "Accept" recommendation.

Reviewer #1: (No Response)

Reviewer #2: All comments have been addressed

2. Is the manuscript technically sound, and do the data support the conclusions?

Reviewer #1: Partly

Reviewer #2: Yes

3. Has the statistical analysis been performed appropriately and rigorously? 

Reviewer #1: No

Reviewer #2: Yes

4. Have the authors made all data underlying the findings in their manuscript fully available?

Reviewer #1: Yes

Reviewer #2: Yes

5. Is the manuscript presented in an intelligible fashion and written in standard English?

Reviewer #1: Yes

Reviewer #2: Yes

6. Review Comments to the Author

Reviewer #1: The explanation that your effort is ‘descriptive’ instead of ‘causal’ is not sufficient to justify your modelling choices. You are still using inferential models which are subject to spatial bias and underestimation due to the lack of all needed predictors. Moran’s I test should be done and spatial modelling done depending on those results. Even with a large dataset like yours, this should be doable on R. Also, you should include population density and some measure of accessibility.

Also, I am well aware of the geography in which you are focused. My reservation is what is to be gained by such a large-scaled focus when this data is capable of so much more. As it stands, that is not enough of a reason to give a negative recommendation, however, more should be done to sell the focus on such a scale. Why should we care about census regions when so much of the variation is happening within these regions?

Reviewer #2: (No Response)

7. PLOS authors have the option to publish the peer review history of their article (what does this mean?). If published, this will include your full peer review and any attached files.

Reviewer #1: No

Reviewer #2: No

---

## [Author Response · Author response to Decision Letter 1]

29 Sep 2021

Please see the response to reviewers file uploaded.

---

## [Decision Letter · Decision Letter 2]

25 Oct 2021

U.S. Regional Differences in Physical Distancing: Evaluating Racial and Socioeconomic Divides During the COVID-19 Pandemic

PONE-D-21-13855R2

Dear Dr. Zang,

We’re pleased to inform you that your manuscript has been judged scientifically suitable for publication and will be formally accepted for publication once it meets all outstanding technical requirements.

Kind regards,

Nickolas D. Zaller

Academic Editor

PLOS ONE

Additional Editor Comments (optional):

Reviewers' comments:

Reviewer's Responses to Questions

**Comments to the Author**

1. If the authors have adequately addressed your comments raised in a previous round of review and you feel that this manuscript is now acceptable for publication, you may indicate that here to bypass the “Comments to the Author” section, enter your conflict of interest statement in the “Confidential to Editor” section, and submit your "Accept" recommendation.

Reviewer #1: All comments have been addressed

2. Is the manuscript technically sound, and do the data support the conclusions?

Reviewer #1: Yes

3. Has the statistical analysis been performed appropriately and rigorously? 

Reviewer #1: Yes

4. Have the authors made all data underlying the findings in their manuscript fully available?

Reviewer #1: Yes

5. Is the manuscript presented in an intelligible fashion and written in standard English?

Reviewer #1: Yes

6. Review Comments to the Author

Reviewer #1: (No Response)

7. PLOS authors have the option to publish the peer review history of their article (what does this mean?). If published, this will include your full peer review and any attached files.

Reviewer #1: No

---

## [Editor Report · Acceptance letter]

17 Nov 2021

PONE-D-21-13855R2 

U.S. regional differences in physical distancing: Evaluating racial and socioeconomic divides during the COVID-19 pandemic 

Dear Dr. Zang:

I'm pleased to inform you that your manuscript has been deemed suitable for publication in PLOS ONE. Congratulations! Your manuscript is now with our production department. 

Kind regards, 

on behalf of

Dr. Nickolas D. Zaller 

Academic Editor

PLOS ONE